# Tetrahydrocurcumin Has Similar Anti-Amyloid Properties as Curcumin:  In Vitro Comparative Structure-Activity Studies

**DOI:** 10.3390/antiox10101592

**Published:** 2021-10-11

**Authors:** Panchanan Maiti, Jayeeta Manna, Joshua Thammathong, Bobbi Evans, Kshatresh Dutta Dubey, Souvik Banerjee, Gary L. Dunbar

**Affiliations:** 1Field Neurosciences Institute, Ascension St. Mary’s Hospital, Saginaw, MI 48604, USA; jayeeta.manna@ascension.org; 2Field Neurosciences Institute Laboratory for Restorative Neurology, Central Michigan University, Mt. Pleasant, MI 48859, USA; 3Program in Neuroscience, Central Michigan University, Mt. Pleasant, MI 48859, USA; 4Department of Psychology, Central Michigan University, Mt. Pleasant, MI 48859, USA; 5Brain Research Laboratory, College of Health and Human Services, Saginaw Valley State University, Saginaw, MI 48604, USA; 6Department of Physical Sciences, College of Science, Technology, Engineering, and Mathematics, University of Arkansas Fort Smith, Fort Smith, AR 72913, USA; jthamm01@g.uafs.edu (J.T.); bevans03@g.uafs.edu (B.E.); Souvik.Banerjee@uafs.edu (S.B.); 7Department of Chemistry and Center for Informatics, School of Natural Sciences, Shiv Nadar University, Delhi-NCR, Gautam Buddha Nagar 201314, India; kshatresh.dubey@snu.edu.in

**Keywords:** Alzheimer’s disease, amyloid beta protein, neurodegeneration, curcumin, molecular docking, MD simulation, docking site, binding energy, structure-activity study

## Abstract

Despite its potent anti-amyloid properties, the utility of curcumin (Cur) for the treatment of Alzheimer’s disease (AD) is limited due to its low bioavailability. Tetrahydrocurcumin (THC), a more stable metabolite has been found in Cur-treated tissues. We compared the anti-amyloid and neuroprotective properties of curcumin, bisdemethoxycurcumin (BDMC), demethoxycurcumin (DMC) and THC using molecular docking/dynamics, in-silico and in vitro studies. We measured the binding affinity, H-bonding capabilities of these compounds with amyloid beta protein (Aβ). Dot blot assays, photo-induced cross linking of unmodified protein (PICUP) and transmission electron microscopy (TEM) were performed to monitor the Aβ aggregation inhibition using these compounds. Neuroprotective effects of these derivatives were evaluated in N2a, CHO and SH-SY5Y cells using Aβ42 (10 µM) as a toxin. Finally, Aβ-binding capabilities were compared in the brain tissue derived from the 5× FAD mouse model of AD. We observed that THC had similar binding capability and Aβ aggregation inhibition such as keto/enol Cur and it was greater than BDMC and DMC. All these derivatives showed a similar degree of neuroprotection in vitro and labeled Aβ-plaques ex vivo. Overall, ECur and THC showed greater anti-amyloid properties than other derivatives. Therefore, THC, a more stable and bioavailable metabolite may provide greater therapeutic efficacy in AD than other turmeric derivatives.

## 1. Introduction

Progressive synaptic damage, neuronal loss and increased neuroinflammation, along with memory impairment, are the main characteristic features in Alzheimer’s disease (AD) [1,2]. Major pathologies that are associated with these symptoms are related to accumulations of denatured, misfolded or mutated amyloid beta proteins (Aβ) and hyper-phosphorylated tau inside and outside of neurons (AD) [3,4,5]. Even though Aβ exists in brain tissue as a metastable structure, plenty of experimental evidence suggests that the soluble form of Aβ is more neurotoxic than the insoluble one [6]. Therefore, targeting neurotoxic species of Aβ and inhibiting their aggregation might be one of the primary goals to prevent or delay the progression of this disease [7]. Numerous small molecules, synthetic drugs and antioxidant molecules have been tested to inhibit Aβ aggregation, many of them have limited bioavailability, with some side-effects. For instance, some of them only delivered a temporary suppression of the AD progression. Moreover, many of these small-molecules or drugs can inhibit Aβ aggregation in vitro and in animal models of AD, but do not show any neurobehavioral improvement in human AD patients. Therefore, finding better anti-amyloid compounds against AD is critically important.

Due to the unsatisfactory outcomes of different anti-amyloid compounds, many natural anti-amyloid compounds have been investigated for treating AD pathologies [8]. Recently, as a potent anti-amyloid, anti-inflammatory natural polyphenol, curcumin (Cur) has shown promising effects against AD progression [9,10,11]. Curcuminoids are derived from the root of the herb, *Curcuma longa.* It includes curcumin (Cur), bisdemethoxycurcumin (BDMC) and demethoxycurcumin (DMC) account for 2–9% of the active compounds of turmeric, depending on the species, cultivation and processing conditions of the rhizome [12,13]. Most of these Cur derivatives are bright, yellow-colored pigment. They account for 60–80% of the main coloring substance in whole turmeric extract [12]. Recent pharmacokinetic and pharmacodynamic studies revealed that Cur produces a more stable, water-soluble, metabolic by-product, called tetrahydrocurcumin (THC) which lacks α, β-unsaturated carbonyl moiety and is white in color [14]. Experimental evidence has shown that these derivatives are non-toxic even at higher doses (2–4 g/day) when taken orally [12]. The hydrophobic nature of these Cur-derivatives facilitates their ability to cross the blood-brain barrier (BBB) efficiently, which increases their binding with Aβ plaques [9,15,16]. Their preferential binding to the β-sheet structures of Aβ confirms that their binding is not dependent upon a specific amino acid sequence of the proteins, but rather indicates that it is conformation-dependent [17]. Interestingly, these Cur-derivative can interfere with Aβ oligomerization better than ibuprofen and naproxen, the classical anti-inflammatory drugs [18].

Even though the anti-amyloid activities of Cur have been extensively studied [11,19,20], less data are available about the anti-amyloid properties of other derivatives of turmeric extract, such BDMC, DMC and water-soluble and stable Cur metabolites, such as THC. Numerous experimental and theoretical investigations have been conducted using different anti-amyloid compounds with Aβ, using high-throughput screening. Computer-aided drug design plays an important role in development of therapeutically relevant small molecules. Molecular and the coarse grain modeling, and interaction field theories, such as quantum and molecular modeling, including quantitative structure-activity relation (QSAR), virtual screening and calculation of pharmacokinetic characteristics are used to determine the ligand binding sites in molecule and/ or protein target structures. Most of these concepts are obtained from using the well-known laws of classical mechanics theory, by approximating the treatment of molecules in terms of atoms, which are thought of as charged spheres linked by springs, disregarding the contribution of electrons and appropriately using the force field (FF) functions by approximating the potential energy contribution. It can be thought of as the totality of bonded intramolecular-nonbonded energy contribution due to van der Waals and Coulomb electrostatic forces. The fundamental idea behind such (usually trigonometric) functions includes bond bending-stretching by harmonic and torsional potential [21,22,23].

Molecular dynamics simulation (MDS) is a computational technique to study dynamic behavior of molecular systems in terms of time, treating all the objects such as ligands, protein, molecules as flexible in the simulation box within an explicit aqueous environment. Molecular dynamics calculate the movements of atoms over time by the integration of Newton’s equations of motions, known as classical mechanics. Such computational chemistry and statistical mechanics can be used to forecast physicochemical properties, binding energies, binding modes, torsional angle interactions and a variety of useful data can provide drug-molecule interaction discoveries and optimization. Recently, the molecular modeling and simulation has been used to study the various molecular docking by choosing a variety of curcuminoid derivatives with Aβ inhibitors. However, very little information is available about comparative in vitro and in vivo studies of these derivatives with Aβ structure-activity [17]. Therefore, here we employed the molecular modeling and simulation to study the various molecular dockings by choosing a variety of curcuminoid derivatives with Aβ inhibitors in explicit solvent environments. In addition, the molecular mechanisms of inhibition of Aβ aggregation have not been extensively explored with these Cur-derivatives. Most importantly, most of these Cur-derivatives are insoluble in body fluids and eliminated within a few hours after systemic administration, except THC, has been found in systemic circulation as a stable and soluble metabolic by-product of Cur-derivatives [24]. These findings prompted us to explore further the comparative efficacy of these Cur-derivatives as Aβ aggregation inhibitors by comparing their interaction with Aβ in both in vitro and in vivo AD models.

The aims of the present study were to investigate the most potent Aβ aggregation inhibitors within Cur derivatives and Cur metabolites. With the aid of molecular dynamics simulation, in vitro and in vivo assays, we have investigated the structure-function relationship of different Cur-derivatives with Aβ by measuring binding energy, torsion angle and hydrogen-bond formation. In addition, we estimated the number of Cur-derivatives required for Aβ aggregation inhibition. Furthermore, we investigated comparative neuroprotective effects of these derivatives in vitro, using Aβ42 as a neurotoxin. Apoptotic markers, cell-survival markers and molecular chaperone levels were investigated after treatment of neuronal cells with Cur and THC. We also compared the binding capabilities of these compounds with amyloid plaques in brain tissue of 5× FAD mice, a mouse model of AD. Our studies showed that all Cur derivatives have affinity to bind with Aβ, but ECur had greater binding energy. All these derivatives showed neuroprotection by inhibiting Aβ aggregation. These data suggested that the THC has potent antiamyloid properties similar to Cur and better than DMC and BDMC.

## 2. Materials and Methods

### 2.1. Chemicals

The list of the chemicals used in this study and their sources are documented as Appendix A.

### 2.2. Molecular Docking and Molecular Dynamic Studies

The structures of two peptides selected for this study are Aβ-peptide1-40 (PDB: 2m4j) and Aβ-peptide1-42 (PDB: 1iyt). These two peptides were downloaded from the Protein Data Bank in pdb format, visualized with the Discovery Studio Visualizer Chain A were selected for both peptides and water molecules were deleted and saved in pdb format for subsequent studies. 

Molecular Docking. Molecular docking was performed using PyRx/AutoDock Vina software and visualized with Discovery Studio Visualizer. The chain A of AB40 (2m4j) and AB42 (1iyt) was selected, water and ions were deleted, missing atoms, if any, were replaced, and polar hydrogens were added. Next, partial Kollman charges were calculated for these peptides and the peptides were set to be rigid. The peptide structures were subjected to restrained energy minimization, adopting GROMOS96 43B1 force field. The energy minimization was performed using Swiss PDB Viewer 4.1.0. These steps were performed using the AutodockTools-1.5.6 and the Swiss PDB Viewer 4.1.0. The size of the grid box was set to 46 × 28 × 27Å for Aβ40 peptide and 37 × 27 × 55 Å for the Aβ42 peptide. Energy of the five Cur derivatives were minimized adopting Merck Molecular Force Field (MMFF94). Docking of each curcumin derivative produced a score for filtering, as well as a model for further validation by molecular dynamic (MD) simulations.

Molecular Dynamics Simulation. Stability of binding of selected curcumin derivatives in an explicit solvent was determined by molecular dynamics (MD) using AMBER 20 MD packages with Amber ff14SB force field. System preparation was carried out using the Leap Module of Amber 20. Forcefield foe the Kcur and THC inhibitors were prepared using the Generalized Amber Forcefield (gaff2) using antechamber module of Amber20. A few Na+ ions were added into the protein surface to neutralize the total charge of the system. Each protein-drug complex was solvated in a triclinic box using the TIP3P water model. The systems were subjected to 5000 steps of steepest descent energy minimization and equilibration under constant NVT (1 ns) and constant NPT (2 ns). During the equilibration, position restraints were applied to both protein and curcumin molecules. The energy minimized structures underwent simulation at constant temperature (310 K) and pressure (1 atm) by the Langevin and Langevin piston methods using periodic boundary conditions. A 50-ns production run was then carried out for each equilibrated system at constant NPT without restraints. The atomic coordinates of simulated structures were evenly recorded at every 10 ps for further analysis. Furthermore, we performed two more replicas for each complex starting from the random velocity. Thereafter we seamed all trajectory to a single MD trajectory and calculated the most populated trajectory using clustering methods. The binding energy by MMPBSA and the final snapshots as shown in later sections are calculated for the most populated trajectory which is a statistically more accurate way to present the MD results [25].

### 2.3. Inhibition of Aβ-Aggregation by Cur/Turmeric Derivatives Using Dot-Blot Assays

To investigate the overall effects of Cur-derivatives on Aβ-aggregation we utilized dot blot assays, as described previously [26]. Synthesized Aβ42 peptide was used for this purpose. For complete monomerization, the Aβ42 peptide was first dissolved in 1,1,1,3,3,3-hexafluoro isopropanol (HFIP), followed by a sonication for 1 min, and incubation additional 30 min at room temperature. The HFIP was evaporated by putting the tubes under a laminar hood for overnight. Next day, the tubes containing peptide films were stored at −20 °C until use. Prior to experiment, each peptide film was dissolved in 60 mM NaOH (final concentration 6 mM) and diluted with 20 mM sodium phosphate buffer (pH 7.4, 0.025% NaN3) to acquire the desired peptide concentration (10 μM). Then 20 μL of peptide solution (10 μM) was taken in Eppendorf tube and incubated in the presence or absence of Cur, BDMC, BMC and THC (1 µM each) for 24–48 h at 37 °C with gentle shaking (200 rpm). For Cur and THC dose-dependent studies different concentrations of Cur or THC (in µM: 10, 1, 0.1 and 0.01) were incubated at 37 °C, for 24–48 h under the shaker (200 rpm). For dot blot assays, about 10 μL of peptide solution was placed on nitrocellulose membrane (Bio-Rad, CA, USA) and allowed to dry. After drying, the nitrocellulose membrane containing dried peptide was washed with PBS for 2 min, followed by a blocking for 1 h at room temperature with 5% nonfat milk powder which was dissolved in TBS-Tween-20 (TBS-T). After incubation, the excess blocking solution was discarded and the membrane was incubated with 6E10, Aβ-oligomer (A11) and fibril specific (OC) rabbit polyclonal antibodies (1: 1000, diluted in same blocking solution) for overnight at 4 °C under gentle shaking. Next day, the blot was washed three times, 10 min each and then incubated with appropriate secondary antibodies conjugated with horseradish-peroxidase (HRP) enzyme (1: 20,000, Santa Cruz Biotech, CA, USA) for 1 h at room temperature. The blot was developed with Super Signal chemiluminescent detection reagent. The dot blots were scanned using a gel documentation system (Bio-Rad, CA, USA), and the optical density of each dot was measured using Image-J software (http://imagej.nih.gov/ij (accessed on 5 June 2021)).

### 2.4. Photo-Induced Cross-Linking of Unmodified Protein (PICUP)

Due to the metastable nature of Aβ, it is difficult to measure quantitatively using classical methods, such as electrophoresis, chromatography, fluorescence or dynamic light scattering. PICUP stabilizes oligomer populations by covalent crosslinking and provides snapshots of the oligomer size distributions that existed before cross-linking as described previously [27,28,29,30,31]. Briefly, ~100 μg of lyophilized Aβ42 peptide was pre-treated with pre-chill 1,1,1,3,3,3-hexafluoro-2-propanol (HFIP) to obtain homogeneous, aggregate-free preparations inside a fume hood. The peptide solution was sonicated in a water-bath for 5 min on ice, vortexed gently and incubated the tubes for 30 min at room temperature. The HFIP was evaporated by placing the tube open in a rack and covering them with a large sheet of Kim-wipes to prevent dust and allowed it for overnight at 37 °C water incubator.

Peptide solubilization. The HFIP-treated peptide was first dissolved in 10% NaOH solution (60 mM, final NaOH solution ~6 mM) and sonicated for 1 min on ice. Then, 40% deionized water was added and vortexed, followed by 50% of 20 mM sodium phosphate buffer (pH 7.4) which was added to acquire 40 µM final peptide concentration. The peptide solution was then aliquoted with equal amount into 0.6 mL low absorbent Eppendorf tubes and incubated at 37 °C with gentle shaking (200 rpm) for 24–48 h [27,28,29,30,31].

Drug treatment. Cur, BDMC, DMC and THC were dissolved in methanol and diluted in a 20 mM sodium phosphate buffer (pH 7.4). The final concentration of each compound was 1 µM. For effects of different doses of Cur and THC on Aβ42 aggregation, 10-, 1-, 0.1- and 0.001- µM concentrations of Cur or THC were used.

Photo-induced cross-linking reaction preparation. Before cross-linking reactions, two reagents ammonium persulphate and tris(2,2-bipyridyl) dichlororuthenium (II) hexahydrate (RuBpy), were prepared freshly. Ammonium persulphate (APS, 20 mM) solution was dissolved in 10 mM sodium phosphate solution (pH 7.4). Then, 1 mM solution of RuBpy was prepared by dissolving in 10 mM sodium phosphate (pH 7.4). The detailed amount of different reagents for cross-linking reaction is documented in Appendix A [27,28,29,30,31]. After adding all these reagents, the camera shutter-delay was adjusted to 1 sec, and the camera shutter was loaded and irradiated for 1 s to initiate cross-linking. A higher irradiation times, extensive radical reactions may cause protein degradation. Therefore, 1 s time was set for optimum reaction. The ratio of different reagents use for a typical PICUP reaction is documented in Appendix A, as described previously [27,28,29].

Sodium-dodecyl-sulphate-polyacrylamide gel electrophoresis. After cross-linking, the peptide samples were mixed with a 5× SDS-Sample buffer. Then the samples were heated at 100 °C for 5 min, allowed to cool at room temperature, and loaded on 4–20% tris-glycine polyacrylamide gel and run at 100 V in XCell SureLock Mini-Cell system (Invitrogen).

Silver stain. After running the SDS-PAGE, the gel was stained by Pierce Silver Stain solutions, as per the instruction manual [29,30,31]. Briefly, the gel was washed with ultrapure water for two times, 5 in each, followed by fixing with 30% ethanol with 10% acetic acid solution two times, 15 min each. Then, the gel was washed with 10% ethanol for two times, 5 min each and followed by two washes with ultrapure water for 5 min each. The gel was sensitized with a working sensitizer solution (50 µL Sensitizer with 25 mL of ultrapure water) for 1 min and then washed by ultrapure water two times, 5 min each. Then, the gel was stained with a working staining solution (0.5 mL Enhancer with 25 mL stain) for 30 min under the shaker at room temperature, followed by two washes at 20 s each. The color was developed with a working Developer solution (0.5 mL enhancer with 25 mL developer). The reaction was stopped with 5% acetic acid solution in ultrapure water for 10 min. The gel was dried with gel drying solution (40% methanol, 10 glycerol, 7.5% acetic acid and 42.5% water). The gel was scanned and optical density of lower order oligomers (4–25 kDa), higher order oligomers (30–80 kDa) and fibrils (above 80 kDa) were measured using Image J software.

### 2.5. Transmission Electron Microscopy (TEM)

To determine whether Cur/turmeric derivatives inhibit Aβ42 aggregation, the morphological changes of Aβ42 was studied by TEM after treatment with Cur/turmeric derivatives, as described previously [30,31]. Briefly, HFIP-treated Aβ42 (10 µM) film was dissolved in 60 mM NaOH (final concentration 6 mM) and diluted with Tris-buffer saline (TBS, 0.1M, pH 7.4, 0.025% NaN3) to obtain a 10-µM concentration of Aβ42 solution. Then 20 μL of peptide solution (10 µM) was taken in sterile Eppendorf tube and incubated in the presence or absence of Cur and THC (1 µM) for 24 h at 37 °C with gentle shaking (200 revolution per min). On the next day, aliquots (8 µL) of peptide solution were applied on a glow-discharges carbon-coated copper grid (grid size 300 mesh × 83 μm pitch) and allowed to dry for 30 min at room temperature. Then the peptide was fixed with 5 μL of 2.5% glutaraldehyde for 4 min and stained with 5 μL of 1% uranyl acetate and lead acetate (filtered through a 0.2-μm syringe filter) for 3 min, followed by the grid being dipped into a drop of double-distilled H_2_O to rinse. The solution was wicked off using Whatman’s filter paper and the grids were air-dried. The morphology was visualized using a transmission electron microscope (JEOL JEM-1400).

### 2.6. Cell Culture

Mouse neuroblastoma cells (N2a), and human cortical cell lines (SH-SY5Y) were used to investigate neurotoxicity for this study. Briefly, N2a, CHO and SH-SY5Y cells were grown in minimum essential medium or DMEM: F12K media, with both cultures containing 10% heat-inactivated fetal bovine serum (FBS) and penicillin/streptomycin (1 μg/mL). The cultures were maintained at 37 °C in a humidified atmosphere at 5% CO_2_. Prior to the experiment, the cells were grown in 96-well plates with either fresh MEM or DMEM/F12K, lacking growth factors [30,31].

### 2.7. Treatment of Different Cur/Turmeric Derivatives

The Cur, BDMC, BMC and THC were dissolved in methanol and then diluted with Dulbecco Phosphate Buffer Saline (DPBS), as described previously (Maiti et al. 2016). The final methanol concentration in the treated solution was ≤1% (*v*/*v*. Initially, different doses of Cur have been studied and based on the experimental results, we found that 1 μM is more potent than any other concentration [32]. Therefore, we used this dose of Cur/turmeric derivatives in rest of the experiments.

### 2.8. Cell Viability by MTT Assay

To evaluate the effect of different Cur-derivatives on neuronal cells, we performed a cell viability assay using MTT [3-(4,5-dimethylthiazol-2-yl)-2,5-diphenyltetrazolium bromide] as a reducing agent. The detailed protocol has been described previously [30,31,32]. Briefly, mouse and human neuroblastoma cells (N2a and SH-SY5Y) were grown with DMEM:F12K media in 96-well plates (COSTAR, Corning, NY, USA) at a density of 3 × 10^5^ cells/mL, without growth factors for overnight. Next day, the cells were treated with freshly prepared Aβ42 (10 μM) as a toxin and simultaneously different Cur-derivatives were added (final concentration = 1 μM) to the treatment groups and incubated for 24 h. In our previous experiment, with low concentration of Aβ42, such as 100 nM or 1 µM, we observed 5–10% cell death. In order to show neuroprotective effect of Cur-derivative, we need to acquire an optimum cell death (may be 30–40%) and with several experiment we have shown that 10 µM of Aβ42 showed 30–40% cell death, therefore, we performed our all the neurotoxicity assays using 10 µM concentration of Aβ42.

The concentration of Cur-derivatives was selected based on our previously published experiment [30,31,32]. In this case, 24 h of Aβ42 and Cur-derivatives treatment, 15 μL of MTT (12 mM) was added to each well and incubated for 4 h at 37 °C. Following incubation, 100 µL of the stop solution (20% SDS in 80% ethyl alcohol) was added to each well to solubilize the MTT crystal and then the plate was kept undisturbed for overnight at room temperature. Next day, the optical density of each well was measured at 570 nm using a Synergy plate reader (Bio-TEK instruments, Winooski, VT). Percentage of cell viability was calculated from each experiment using vehicle treated cells as control and three independent experiments (8 wells per condition) were performed to acquire the average value of each group.

### 2.9. Western Blot

The SH-SY5Y cells were grown in DMEM: F12K media with 10% FBS and 1% streptomycin/penicillin. Prior to treatment the cells were grown in DMEM: F12K without growth factor for 24 h. Then, the cells were treated with Aβ42 (10 µM) in presence and absence of different concentrations of Cur and THC for 24 h. After 24 h of treatment, the cells were scraped, and pellet was lysed by lysis buffer and supernatant was collected and aliquoted in 20 µL in 0.2-mL PCR tube and stored at −80 °C until further use. Total protein was measured by a BCA kit (ThermoFisher). The protein samples were added with equal amount of 2xSDS-sample buffer (125 mM Tris-HCl, pH 6.8, 4% sodium dodecyl sulfate, 20% glycerol and 10% 2-mercaptoethanol) and boiled for 5 min. Approximately, 100 µg of protein per lane was loaded and electrophoresed on 4–20% tris-glycine gel and transferred to PVDF membrane (Millipore, Bedford, MA, USA). The blot was probed with rabbit polyclonal anti-caspase 3, Akt antibodies (1:1000) and incubated overnight at 4 °C under the shaker. On the next day, the blot was probed with anti-rabbit secondary, antibody conjugated with horseradish peroxidase (HRP) and signal was developed with SuperSignal West Femto Chemiluminescent HRP-Substrate (Thermo Fisher, Catalog no: 34096). Similarly, SH-SY5Y cells were treated with different concentrations (in µM: 10, 1, 0.1 and 0.01) of THC and protein was extracted, and the SDS-PAGE was run and probed with HSP90 and HSP70 antibodies. The same blot was also probed with anti-beta tubulin antibody for loading control. The relative optical density was measured using Image-J software (https://imagej.nih.gov/ij/ (accessed on 5 June 2021) [32,33,34].

### 2.10. Animals

To check the binding capabilities of different Cur-derivatives with amyloid plaques, 5× familial Alzheimer’s disease (5× FAD) mice brain tissue and age-matched wild-type mice were used, as described previously [33,34]. Briefly, B6SJL-Tg mice (APPSwFlLon, 1136799Vas/J) which overexpressed human amyloid precursor protein (APP) with three mutations [Swedish (K670N, M671L), Florida (I716V) and London (V717I)] and two mutations on presenilin (PS) genes (PSEN1*M146L*L286V)] [7,8]. By the age of 12 month, these mice developed a plenty amount of Aβ-plaques in the cortex and hippocampus. These mice were housed at the neuroscience vivarium of Saginaw Valley State University. The standard housing conditions including temperature at 22 °C and ad libitum access of food and water with 12-h light/12-h dark, reverse-light cycle was maintained. The transgenic characteristics of 5× FAD mice were confirmed by genotyping using polymerase chain reaction (PCR) from the tail snip at the age of 3 weeks [33,34]. The Institutional Animal Care and Use Committee of the Saginaw Valley State University approved this study (IACUC no: 1513829-1 and 1645929-2). Animal sacrificed was performed under the overdose of fetal plus, and all efforts were made to minimize their suffering.

### 2.11. Tissue Processing and Aβ Plaque Labelling by Cur/Turmeric Derivatives

The detail protocol for Aβ plaque labeling in 5× FAD brain tissue using Cur was described previously [16,33,35]. Briefly, all mice were deeply anaesthetized with an overdose of sodium pentobarbital (intraperitoneally), and transcardially perfused with 0.1 M PBS, followed by 4% paraformaldehyde (diluted in 0.1 M PBS at pH 7.4) to fix the brains.

#### 2.11.1. Aβ Labelling Using the Cryostat Section

For labelling Aβ plaques from 5× FAD brain tissue, the perfused whole brain tissue was treated with graded sucrose solutions, followed by cryostat sectioning and labelled with Cur-derivatives, as described previously [16,33,35]. Briefly, brain sections (40 µm thick, coronal) were washed with freshly prepared PBS, three-times, 2 min each, followed by dip to 70% alcohol and stained with Cur-derivatives (Stock Cur derivatives were dissolved in methanol and diluted with 70% alcohol to acquire 1 µM final concentration) for 10 min at room temperature in dark with gentle shaking. After incubation, the staining materials were discarded, the sections were washed with 70% ethanol, thrice, 2 min each and finally placed them on a poly-L-lysine coated glass slide. The tissue was mounted with a coverslip using organic mounting media, (distyrene plasticizer xylene; DPX). When dried, the sections were visualized under a fluorescence microscope (Leica, Wetzlar, Germany) at a 20× objective (total magnification = 200×) using 480/550 nm excitation/emission filters [16,33,35].

#### 2.11.2. Aβ Labelling Using the Paraffin Section

For paraffin sections, the brain tissue was dehydrated with graded alcohols (50%, 70%, 90%) for 2 h each, followed by 100% alcohol 2× for 1 h each), and then with xylene 2× for 1 h each) at room temperature. Tissue was penetrated with xylene-paraffin (1:1) 2× for 1 h at 56 °C and was immersed in melted paraffin (56 °C) for 4−6 h. Five-micron thick coronal sections were cut using a rotary microtome at room temperature and collected on a poly-lysine-treated glass slide and stored at room temperature for several days. The sections were deparaffinized with xylene, two-times, 5 min each, at room temperature. They were rehydrated with graded alcohol solutions (100%, 80%, 70%, 50% for 1 min each) and with distilled water, twice, 5 min each, at room temperature. Then the sections were stained with Cur/turmeric derivatives (1 µM) for 10 min at room temperature in the dark, shaking at 150 rpm and washed with 70%, 90% and 100% alcohol for 2 min each., cleared with xylene, twice, 5 min each and cover slipped with DPX. Sections were visualized under a fluorescence microscope (Leica, Wetzlar, Germany) using 20× objective (total magnification = 200×), as mentioned above. Bright, green fluorescent signal was considered as Aβ plaques, which was confirmed by co-labeling with Aβ-specific antibody (6E10), as described previously [16,33,35].

### 2.12. Statistical Analysis

The analyzed data for dot blots, PICUP, cell viability and Western blot experiments were expressed as mean ± SEM. For the statistical analyses, one-way analysis of variance (ANOVA) was used. A Tukey HSD (honestly significant difference) post hoc test was performed whenever found significant difference among the groups. The probability value was set to 0.05 for considering statistically significant difference.

## 3. Results

### 3.1. Curcumin Derivatives Interact with Aβ Similar to Curcumin

Molecular docking studies revealed that keto, enol curcumin, BDMC, DMC and THC all interacted with both Aβ40 and Aβ42 (Figure 1 and Figure 2). We observed the strongest interaction of Aβ with Cur and THC.

Molecular Docking. Molecular docking of five curcumin derivatives demonstrates (Appendix A) that keto-curcumin has stronger binding affinity (−6.3 kcal/mol) for Aβ40 than Aβ42 (−5.4 kcal/mol). However, KCur has overall strong binding affinity for both Aβ40 and Aβ42, given none of the curcumin derivatives have binding affinity greater than −5.5 kcal/mol for the Aβ-42. The more bioavailable THC seems to have poor binding affinity for both Aβ40 and Aβ42. Enol curcumin has exactly equal binding affinity for both Aβ40 and Aβ42. Bis-demethoxycurcumin ranks third for both Aβ40 and 42 with almost equal binding affinity. However, DMC differs significantly in regard to the binding affinity for Aβ40 and 42 with higher binding affinity for Aβ40. Hence, these findings suggest that keto and enol curcumin are more likely to bind both Aβ40 and 42 strongly. In addition, the more bioavailable THC has stronger binding affinity for the Aβ42 over Aβ40.

### 3.2. Binding Affinity of Different Cur Derivatives with Aβ40 and Aβ42

During interaction of different Cur-derivatives with Aβ40 and Aβ42, we measured the binding affinity and the amino acids involved in the interaction. The highest binding affinity was observed in the case of KCur, followed by ECur and the lowest binding energy was noted in the case of THC (KCur > ECur > BDMC > DMC > THC) during interaction with both Aβ40 and Aβ42 (Appendix A). We also compared the H-bond formation of KCur and THC with Aβ40 and Aβ42 and we found that KCur integrated with 20 amino acids, whereas THC interacted with 28 amino acids to form the H-bonds in the case of Aβ40. Whereas, in the case of Aβ42, KCur interacted with 19 amino acids, and THC interacted with 25 amino acids to form the H- bonds (Appendix A).

### 3.3. Molecular Dynamic Studies of Aβ40 and Aβ42 after Interaction with KCur and THC

MD simulation was performed on selected poses of KCur and THC in complex with Aβ40 and Aβ42 (see Figure 3 for initial RMSD and RMSFplots). In all simulations, surprisingly we see a large allosteric conformation changes in both Aβ40 and Aβ42 oligomers induced by the inhibitors. The representative snapshots of the initial and final conformations form the simulations for each protein-inhibitor complex is shown in Figure 4A–D. As can be seen, during the initial stage of the simulations both oligomers possess an unfolded helical structure, however, the binding of inhibitors causes a twist in the helix which, in turn, forms a closed binding pocket to stabilize the inhibitor interaction with oligomers. A closer inspection of the interactions of all four inhibitors reveals that hydrophobic interactions due to Leu1, Phe20, Ile31, Leu34 and Val41 are the key interactions responsible for the stability of the inhibitors while some polar and electrostatic interactions due to Asp7, His14 and Lys16 causes twist in the helical structure. We believe that such inhibitor instigated twisting in the Aβ40 and Aβ42 oligomers might be the root cause the inhibitory function of KCur and THC derivatives. To further explore the binding affinity of these inhibitors, we performed Molecular Mechanical Generalized Boltzmann (MMGBSA) for all four complexes. Binding free energies are shown in Table 1. The energetic calculations also substantiate the key role of the hydrophobic interaction (E_vdw_). The electrostatic interactions are relatively weak, and it doesn’t play significant role in the binding affinity for any of the inhibitors. Interestingly, the THC inhibitors are better binder vis-à-vis Kcur. Among all four complexes KCur binding is weakest with Aβ40.

### 3.4. Bonding Interactions of KCur and THC with Aβ

We also investigated the bonding interaction of KCur and THC with Aβ40 and Aβ42. Figure 4 representing the bonding interactions of KCur and THC with Aβ40 and Aβ42 during the course of simulation. Both of the Cur analogues were predominantly forming hydrophobic, van der Waals and electrostatic interactions with the Aβ42. For KCur, Ala42 seems to be primarily involved in H-bonding with 1% occupancy. For THC, His14, Lys16 and Glu22 are primarily involved in H-bonding with 0.44%, 0.30% and 0.32% occupancy, respectively. Details H-bonding during interaction of KCur and THC with both Aβ40 and Aβ42 are documented at Appendix A.

### 3.5. Cur Derivatives Inhibited Aβ42 Oligomer and Fibril Formation In Vitro

Dot blot assays were performed to assess the inhibitory capability of different Cur and turmeric derivatives. We observed that THC inhibited Aβ42 aggregation after 24 h of incubation, whereas Cur DMC and THC took 48 h to inhibit Aβ42 aggregation, but BDMC was ineffective at both 24- and 48 h (Figure 5A,C). Similarly, Aβ42 fibril formation was significantly inhibited by Cur and turmeric derivatives but greater Aβ42 aggregation inhibition was observed in THC-treated groups (Figure 5A,D). We did not find any significant differences when the blots were probed with 6E10 antibody (Figure 5A,B). In addition, we observed Aβ42 fibril formation was inhibited by both Cur and THC after 24 h of their incubation with Aβ42 (Figure 5E).

### 3.6. Lower Concentrations of Cur and THC Inhibited Aβ42 Oligomers and Fibril Formation Greater than Higher Concentrations

Amyloid peptide 42 was incubated with different concentration of Cur and THC (in µM: 10, 1, 0.1, 0.01) for 24–48 h and then dot blot assays were performed and probed with oligomer and fibril specific antibodies. We observed that 0.01 µM showed greater inhibition of Aβ42 oligomer and fibril in comparison to higher concentrations, such as 10-, 1- and 0.1 µM. In fact, 10 µM of both Cur and THC increased Aβ42 aggregation in both at 24–48 h of incubation (Figure 6A,D–G). When the bolts were probed with 6E10 as loading control, we did not observe any significant differences among the groups (Figure 6A–C).

### 3.7. Photo-Induced Cross-Linking of Unmodified Protein (PICUP) of Aβ42 after Treatment with Cur/Turmeric Derivatives

PICUP was used to quantify the oligomer and fibril formation in vitro after treatment with Cur/turmeric derivatives. The lower molecular weight oligomers (LMWO) were significantly decreased by THC after 48 h of its incubation, whereas other Cur/turmeric derivatives did not inhibit their distribution (Figure 7A,B,D). In addition, we observed a significant decrease of higher molecular weight oligomers (HMWOs) with the treatments of Cur and THC after 24 h of incubation, but not after 48 h of incubation (Figure 7A,B,E). Fibril formation was only inhibited by Cur after 24 h, but not at 48 h of incubation (Figure 7A,B,F).

### 3.8. Photo-Induced Cross-Linking of Unmodified Protein (PICUP) with Aβ42 after Treatment with Different Concentrations of Cur and THC

We compared which concentration of Cur or THC is able to inhibit Aβ42 oligomer and fibril formation using the PICUP method. Lower MW oligomers were significantly inhibited by both Cur and THC in all the concentrations used, except 0.01 µM in both 24- and 48-h of their incubation (Figure 8A–D). Similarly, higher MW oligomers were also inhibited by both Cur and THC at all the concentrations after 24 h of incubation (Figure 8A,B,E,F). After 48 h of incubation only 1- and 0.1-µM of THC inhibited Aβ42 oligomer formation (Figure 8B,F). Similarly, 10- and 1-µM concentrations of both Cur and THC inhibited Aβ42 fibril formation after 24 h of their incubation with Aβ42, whereas after 48 h of incubation, only the 0.01-µM concentration of THC inhibited Aβ42 fibril formation (Figure 8A,B,H).

### 3.9. Curcumin/Turmeric Derivatives Are Equally Effective in Binding and Labeling Amyloid Plaques in 5× FAD Brain Tissue

To investigate whether Cur/turmeric derivatives bind and label Aβ plaque in AD brain tissue, brain tissue from 12-month-old 5× FAD mice were sectioned on a cryostat (40 µM) or paraffin-embedded (5 µM) sectioned with a rotary microtome and stained with all the Cur/turmeric derivatives (1 µM) for 10–20 min. We observed that all these derivatives stained the amyloid plaques in both cryostat and paraffin-embedded tissue sections in a similar fashion. No significant fluorescent intensity difference was observed among all the groups (Figure 9).

### 3.10. Curcumin/Turmeric Derivatives Protected Aβ42-Induced Neurotoxicity In Vitro

To investigate whether Cur/turmeric derivatives have any neuroprotective effects, the N2a and SH-SY5Y cells were treated with Aβ42 (10 µM), followed by Cur/turmeric derivatives (1 µM) for 24 h. We observed that all these Cur/turmeric derivatives, except BDMC in SH-SY5Y protected 50–60% of the cells from Aβ42-induced cell death (Figure 10A,B). Further, our Western blot data suggested that both Cur (1–0.01 µM) and THC (1 µM) significantly increased Akt levels and significantly decreased caspase-3 levels in SH-SY5Y cells treated for 24 h (Figure 10C–E). Further, different concentrations of THC treatment significantly induce HSP90 and HSP70 levels in SH-SY5Y cells in vitro (Figure 10F–H).

## 4. Discussion

Use of curcumin (Cur) for treating Alzheimer’s disease (AD) has gained significant attention because of its potent anti-amyloid and anti-inflammatory properties, as this polyphenol has lower toxicity and less expensive than most other treatment modalities [9,16,18,19,20]. Most researchers focus on the anti-amyloid properties of Cur in turmeric extract; however, it also contains abundant amounts of other polyphenols, especially bisdemethoxycurcumin (BDMC) and demethoxycurcumin (DMC). In addition, these compounds become metabolized in the liver and produce significant amounts of relatively stable, water-soluble metabolite, such as tetrahydrocurcumin (THC). The aims of the present study were to: (i) compare the binding and aggregation inhibition efficiency of Cur, BDMC, DMC and THC with Aβ40 and Aβ42; and (ii) assess the neuroprotective roles of these compounds. To this end, we compared the Aβ binding capability of enol-, and keto-Cur, as well as other derivatives of Cur/turmeric, such as BDMC, DMC and THC in silico and in vitro, using molecular dynamics (MD) and docking studies, dot blot assays, photo-induced cross-linking of unmodified protein (PICUP), transmission electron microscopy and neurotoxicity assays. The MD simulation, docking analysis of AutoDock binding energies and binding interactions of Cur and its derivatives indicate that Cur and THC are the most potent Aβ aggregation inhibitors and confer the most neuroprotective effects, with KCur and THC showing greater affinity toward Aβ than the other Cur/turmeric derivatives.

Curcumin is a diferuloyl methane molecule [1, 7-bis (4-hydroxy-3-methoxyphenyl)-1,6-heptadiene-3,5-dione)] containing two ferulic acid residues joined by a methylene bridge (Figure 1B). It contains three important functional groups: (1) an aromatic o-methoxy phenolic group; (2) an α, β-unsaturated β-diketo moiety; and (3) a seven-carbon linker (Figure 1B). These functional groups have critical roles for the biological properties of Cur [11,36,37,38,39]. However, natural Cur may be less useful for treating neurodegenerative diseases, due to its low solubility in body fluids, rapid degradation after intestinal absorption and its limited bioavailability [11,19,40]. In addition, turmeric also contains other polyphenols which may exert anti-amyloid properties similar to those of Cur. There have been many reports available which described the Cur-binding affinity to Aβ-plaques [9,16,25,35] but less information is available for comparative binding kinetics of Cur/turmeric derivatives and their efficiency for inhibiting Aβ aggregation and promoting neuroprotection. Therefore, in the present study we were interested to compare the Aβ binding and aggregation inhibition capabilities using different in vitro and in silico techniques.

Computer-aided drug designing is an important method for drug discovery, for which molecular docking analysis is one of them. These methods calculate the binding affinities and predicting binding sites, which helps to model and predict structural changes and activity of the interacting molecules [41,42,43]. In the present study, we utilized molecular docking (MD) techniques for the accurate study of binding affinity of Cur and Cur/turmeric derivatives to Aβ40 and Aβ42 to gain new insights for treatment options for AD. We observed that DBMC, DMC, THC all strongly interacted with Aβ40 and Aβ42, similar to keto and enol forms of Cur molecules (Figure 1 and Figure 2). Interestingly, we also observed that most of the compounds have preference to bind at N-terminal sequence to the central hydrophobic area of Aβ, suggesting that N-terminal to the central hydrophobic area are accessible for binding of these molecules to inhibit Aβ aggregation (Figure 1 and Figure 2).

To compare the interactions of these compounds with both Aβ40 and Aβ42, we have calculated the binding or affinity energy. Binding energy computations may prove to be one of the best fruitful methods of assessing free energy of complexes and may reveal to be more pertinent to provide visions into the landscape of interactions rather than estimation and/ or screening process [44,45]. When we compared the binding energy of all these derivatives with Aβ40 and Aβ42, the lowest binding energy was observed in the case of KCur, indicating KCur has stronger binding affinity with both Aβ40 and Aβ42 than the other Cur/turmeric derivatives (Figure 2), followed by ECur, BDMC, DMC and THC (KCur > ECur > BDMC > DMC > THC). This clearly indicates that in the presence of Cur/turmeric derivatives, two Aβ molecules become separated from clumping together. The KCur showed greater affinity to Aβ, because KCur is more hydrophobic, or lipid soluble, which facilitate its penetration to the hydrophobic core of the Aβ aggregates, and thus interfere its further aggregation. In contrast, THC is more stable in hydrophilic environment, including most body fluids than KCur or other Cur-derivatives, which makes it less interactive with hydrophobic residues of Aβ, however, as it is more stable compound which helps it to acquire more time to interact with Aβ, thus facilitate for Aβ aggregation inhibition similar to KCur or other Cur derivatives (Figure 5, Figure 6 and Figure 7).

Further, we have investigated the nature of H-bond formations in absence and presence of different Cur/turmeric derivatives with Aβ. Hydrogen (H)-bonds play an important role in computing the ligand-molecule binding location. It initiates a variety of cellular functions by enabling molecular interactions. The mechanism and the range at which H-bonds control molecular interactions are still an unresolved problem in the biological field, due to bound-bulk water competition and H-bonding process [44,46]. According to Jeffrey’s classification of H-bonds, it has acceptor-donor distances in the range of 2.2–2.5, 2.5–3.2 and 3.2–4.0 Å, indicating as strong-to-moderate weak interactions, respectively [46]. From the MDS study, which incorporates electrostatic interactions, entropy change in explicit solvent and van der Waals interactions, we can predict the H-bond formations of Cur-derivatives with Aβ. The Multiomics Analysis Software (MAS) revealed that such H-bonds depend on the pairing ability of the acceptor-donors. However, solvent entropy changes constantly, due to such pairing and is a complicated task in biological molecules [45]. Previous reports suggest that Cur can form H-bonds with several amino acid residues present in the Aβ, mainly from n-terminal up to 28 bonds, or sometimes with a few C-terminal amino acids [17]. Cur-derivatives exhibited favorable binding energy and interacted with the binding pocket of Aβ, with several amino acids (Table 1 and Appendix A). Our docking study revealed that more amino acids of Aβ interacted with ECur and THC (Figure 4). Recently, Rao and colleague reported that Cur is interacting with the amino acid positions at 22, 24, 26 of Aβ, whereas BDMC with Tyr22, Asp24, Thr26, Thr11, Ala9, Gln8, Phe33, DMC binds with Tyr22, Asp24, Thr26, Thr11, Ala9, Gln8, Glu27, Phe33, Phe34 and THC binds with Thr11, Thr26, Asp24, Tyr22, Ala9, Gln8, Phe32, Phe33 [47]. Even though we did not observe binding in exactly the same way. These discrepancies may be dependent on the software used and also the Aβ isoform, their energy levels and variation of many more experimental parameters. However, our studies and previous reports agreed that, among all these amino acids, Asp, Tyr, Ala, Lys and Thr play greater role for Aβ aggregation and that Cur/turmeric derivatives interacted with these amino acids more favorable [48], suggesting that these Cur/turmeric derivatives have critical roles in Aβ aggregation inhibition [47,49] (Figure 1, Figure 2, Figure 3 and Figure 4, Appendix A).

Interactions and docking results indicate that Cur/turmeric derivatives are strong inhibitors of Aβ protein as they interact with its active site [47,49,50]. We also investigated the interaction between two Aβ42 molecules at distance of 3.99A^◦^ (which is much greater than the distance of a covalent bond) with and without various Cur/turmeric derivatives at 1000 ps s dynamic time. It clearly showed a delay in interacting between the two Aβ peptides in the presence of different Cur/turmeric derivatives. We also, investigated the number of Cur/turmeric derivatives required to have certain effects during Aβ aggregation and we found that minimum 12–18 Cur-molecules are required to inhibit their aggregation, whereas for THC it was 5–6 molecules, indicating THC has greater Aβ42 inhibitory activities than Cur. Similarly, we also found that maximum effect of THC against Aβ was observed to be approximately 12 molecules, and the aggregation of Aβ can be reduced by the critical numbers or amount of Cur or THC molecules more effectively than by other Cur/turmeric derivatives (data not shown).

To demonstrate the anti-aggregation properties of Cur-derivatives, we employed dot blot assays using oligomer and fibril-specific antibodies. After 24–48 h of incubation with all these Cur/turmeric derivatives (at 1 µM), we observed that Cur and THC significantly decreased Aβ42 oligomer formation (Figure 5, Appendix A). These findings are supported by Yang and Colleagues, previously [9]. Interestingly, we observed a greater inhibition of oligomer formation in the case of THC at 48 h of its incubation with Aβ42. Similarly, Aβ42 fibril formation was also significantly inhibited by Cur and THC in both 24 h and 48 h of their incubation and THC showed greater inhibition of fibril formation after 24–48 h of its incubation with Aβ42 (Figure 5, Appendix A) In addition, we also investigated how much concentration of Cur or THC is required to inhibit Aβ42 oligomer and fibril formation. We found that as low as 10 nM concentration of THC inhibited Aβ42 aggregation (Figure 6). To correlate our dot blot assays we performed a photo-induced crosslinking of unmodified protein (PICUP) experiment (Appendix A). Even though this technique was originally developed to study stable protein complexes, the method can be applied to quantitative study of metastable amyloid protein assemblies [27,28,29]. By this technique, we can identify and quantify the oligomers distribution of metastable amyloid proteins, such as Aβ, after SDS-PAGE (Appendix A). Thus, it is an effective technique in understanding oligomer formation in different amyloidogenic diseases. In the present study, we observed that higher MW oligomers (HMWO) formation were inhibited by Cur and THC (Figure 7E), but not by BDMC and DMC, suggesting that both Cur and THC are potent Aβ aggregation inhibitors. Interestingly, we found that THC also significantly inhibited low MW oligomers (LMWO) after 48 h of treatment, suggesting that THC has greater Aβ42 aggregation inhibition abilities than other Cur/turmeric derivatives (Figure 7D). When we treated different concentrations of Cur and THC (in µM: 10, 1, 0.1, 0.01), we observed that most of these concentrations (except the 0.01 µM) inhibited LMWO and HMWO, suggesting that THC has the potential similar to Cur to inhibit Aβ aggregation (Figure 8). Even though Aβ40 and Aβ42 are considered the main toxic species in AD brain, however a significant proportion of AD brain also contain N-terminal truncated species and pyroglutamate-modified Aβ especially, pyroglutamate-modified Aβ (AβpE3), and nitrated Aβ (3NTyr10-Aβ), AβN3(pE) and AβN11(pE) have been demonstrated to be the predominant components among all N-terminal truncated Aβ species in AD brains. AβpE3 serves as a seed for Aβ42 aggregation and might change the binding properties of Cur-derivatives. We have not tested these species of Aβ after treatment with different Cur-derivatives, which need additional experiment.

To check whether these Cur/turmeric derivatives label Aβ plaques in AD brain tissue, we stained the 5× FAD brain tissue with these compounds (at 1 µM) and found that they all bind and label Aβ plaques equally as well as in both cryostat and paraffin-embedded sections (Figure 9). This indicates that all these Cur/turmeric derivatives have preference to interact with Aβ and can be used as amyloid binding dyes. Even though we did not quantify the plaques or fluorescent intensity of the plaques for each compound, tested it appeared that they have equivalent affinities to bind with Aβ plaques. Due to their preferential binding to Aβ plaques, several investigators have used these Cur/turmeric derivatives for amyloid plaque imaging. Recently, Yanagisawa and colleagues reported that the keto form of Cur derivatives (KCur) strongly binds to Aβ oligomers, but not fibrils [51] which strongly corelated with our present study, suggesting KCur has stronger binding to Aβ, whereas THC has similar binding properties with Aβ such as KCur.

To investigate whether these Cur/turmeric derivatives protect Aβ neurotoxicity, the N2a and SH-SY5Y cells were treated with Aβ42 (10 µM) for 24 h in presence and absence of these compounds (at 1 µM) and we found that most of these compounds protected almost 50–60% of the cells from Aβ42-toxin-induced death [32,38]. Among all these compounds, BDMC did not show neuroprotection in SH-SY5Y cells when compared with other Cur/turmeric derivatives, however, the neuroprotective effects exerted by Cur and THC strongly correlated with the dot blot assays, suggesting a strong inhibition of Aβ oligomer formation by Cur and THC. To further investigate the neuroprotective effects of Cur and THC, we measured Akt and Caspase-3 levels of those SH-SY5Y cells that were treated with Aβ42 and we observed that both Cur and THC (at 1 µM) significantly decreased Caspase-3 levels and increased Akt levels, suggesting both these compounds may prevent apoptotic death by decreasing levels of caspase-3 levels and by inducing cell survival markers, such as Akt (Figure 10). Recently, Sandur and colleagues reported that Cur, BDMC, DMC and THC and turmerones differentially regulate anti-inflammatory and anti-proliferative responses (Cur> DMC> BDMC> THC) through a ROS-independent mechanism [38], suggesting THC has similar anti- inflammatory and neuroprotective properties as noted in the case of other Cur-derivatives. In addition, it is speculated that THC would have more neuroprotective effects than other Cur-derivatives as it is more stable and hydrophilic compounds than other Cur-derivatives, which needs additional experiment for confirmation. To confirm our neuroprotective effect, we also investigated the molecular chaperones levels, such as heat shock protein 90 (HSP90) and HSP70, which play a vital role in protein refolding. These proteins are down regulated in protein misfolding diseases, including AD [52,53]. Previously, we have shown that Cur induces chaperone activities in vivo and in vitro [32]. As we observed similar anti-amyloid and neuroprotective effects similar to other Cur-derivatives, we were interested whether THC has any role on induction of molecular chaperones, such as HSPs. Our Western blot data clearly demonstrated that different concentrations of THC induced HSP90 and HSP70 in SH-SY5Y cells, similar to Cur-treatment (Figure 10), suggesting THC has a significant role in protein quality control and Aβ aggregation inhibition [32], as observed in the case of other Cur-derivatives. Even though we did not compare the HSP levels after treatment with BDMC or DMC, along with THC-treated cells, because we were more interested to compare the neuroprotective effect of THC with Cur. Similarly, the molecular mechanisms of induction of HSPs by Cur-derivatives and or by THC is not clear yet, however, our study confirms that THC can induce the HSPs response similar to Cur.

## 5. Conclusions

We found that the capacity of THC to bind and inhibit Aβ aggregation was greater than that of other derivatives. The keto form of Cur (KCur) showed stronger binding to Aβ in comparison to the enol form of Cur (ECur). All Cur/turmeric derivatives (Cur, BDMC, DMC, THC) attenuated Aβ-aggregation significantly. THC also protected Aβ-induced neurotoxicity and chaperone activities in vitro. Overall, the in silico and in vitro data revealed that THC has anti-amyloid and neuroprotective effects that are similar to Cur. Therefore, as a relatively more stable Cur-metabolite, THC is a promising polyphenol which inhibits Aβ aggregation more effectively than other Cur/turmeric derivatives. Further research, especially using other animal models of AD, is needed to confirm these findings and to optimize them for future therapeutic applications.

## Figures and Tables

**Figure 1 antioxidants-10-01592-f001:**
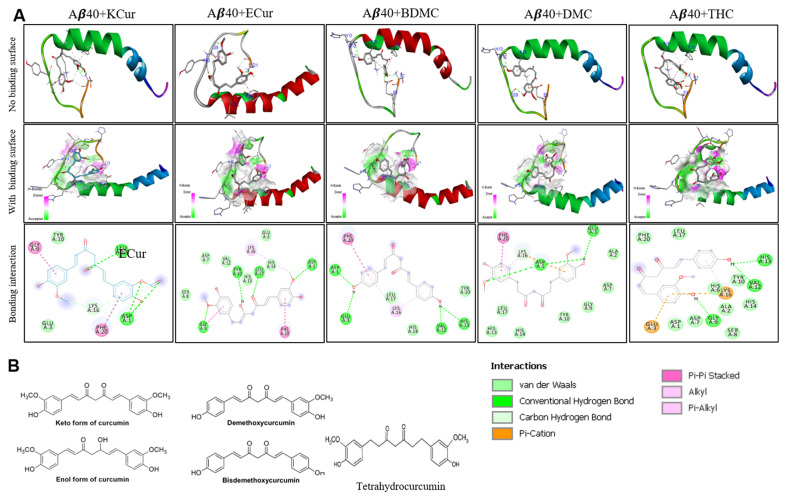
Interactions of Cur and its derivatives with Aβ40. Docking analysis was carried out and the results were investigated using PyRx/AutoDock Vina software and visualized with Discovery Studio Visualizer to further investigate with Aβ40-Cur derivatives. (**A**): The Molecular docking analysis showed that there were strong interactions of BDMC, DMC and THC, similar to enol and keto forms of Cur. The strongest interaction with Aβ40 was observed in the case of KCur and ECur. (**B**): chemical structure of Cur and different Cur derivatives.

**Figure 2 antioxidants-10-01592-f002:**
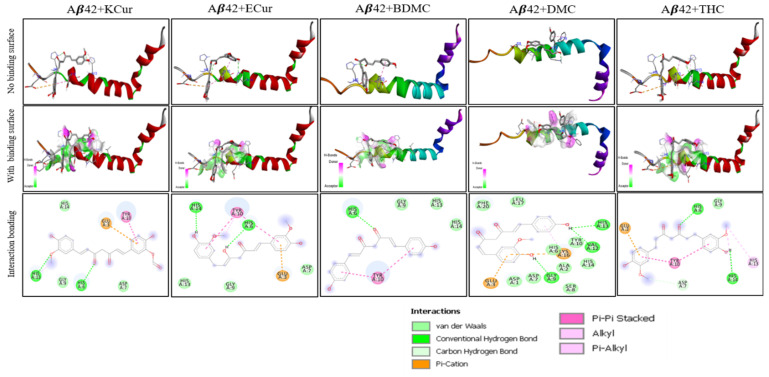
Interactions of Cur and its derivatives with Aβ42. Docking analysis was carried out and the results were investigated using PyRx/AutoDock Vina software and visualized with Discovery Studio Visualizer to further investigate with Aβ40-Cur derivatives. The Molecular docking analysis showed that there were strong interactions of BDMC, DMC and THC, similar to enol and keto forms of Cur. The strongest interaction with Aβ42 was observed in the case of KCur and ECur.

**Figure 3 antioxidants-10-01592-f003:**
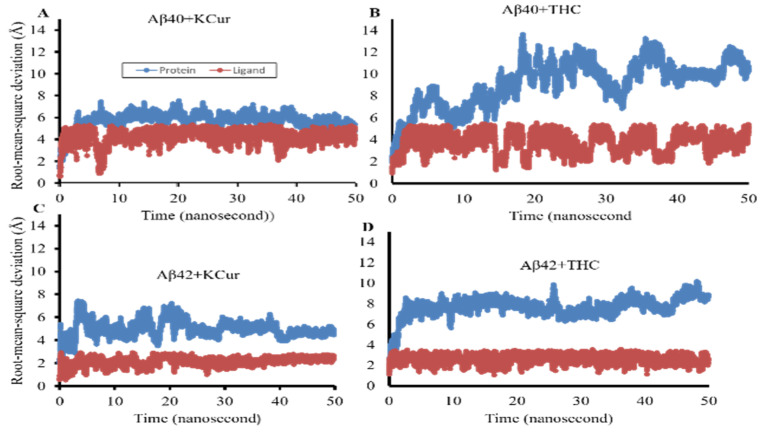
RMSD and RMSF values after interaction of Aβ40 and Aβ42 with KCur and THC. Docking analysis was carried out and the results were investigated using PyRx/AutoDock Vina software and RMSD and RMSF values were with Discovery Studio Visualizer. (**A**–**D**): RMSD values of KCur and THC when interacted with Aβ40 and Aβ42. (**E**–**H**): RMSF values of KCur and THC when interacted with Aβ40 and Aβ42. These findings indicate that KCur has stable complexes with both Aβ40 and 42, while THC can only form stable complexes with Aβ42. RMSD and RMSF are shown for the first 50 ns of the simulations.

**Figure 4 antioxidants-10-01592-f004:**
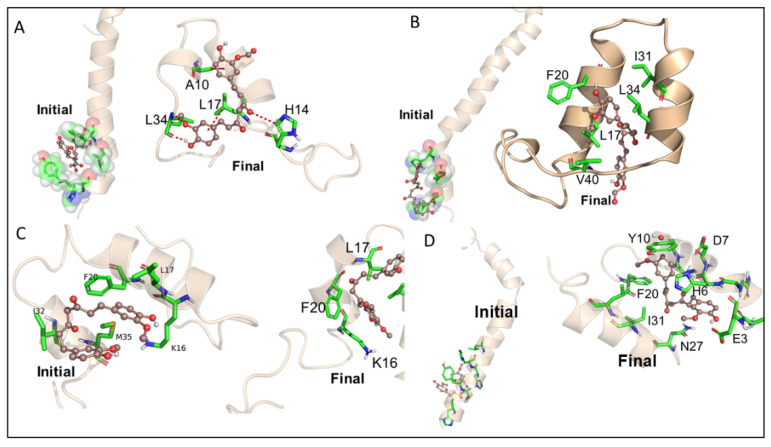
Molecular dynamic studies of Aβ40 and Aβ42 with KCur and THC. Represents the two representative snapshots from the initial and final stages of the MD simulations. These figures clearly represent that THC and KCur induces an allosteric conformational change that causes a twist in both oligomers Aβ42 primarily via hydrophobic interactions and, thus, may be capable of inhibiting the aggregation. THC seems to be interacting over wider surface area of Aβ42 and, thus, may be more effective in inhibiting Aβ42 aggregation, here (**A**) Kcur with Aβ40, (**B**) Kcur with Aβ42, (**C**) THC with Aβ40 and (**D**) THC with Aβ42.

**Figure 5 antioxidants-10-01592-f005:**
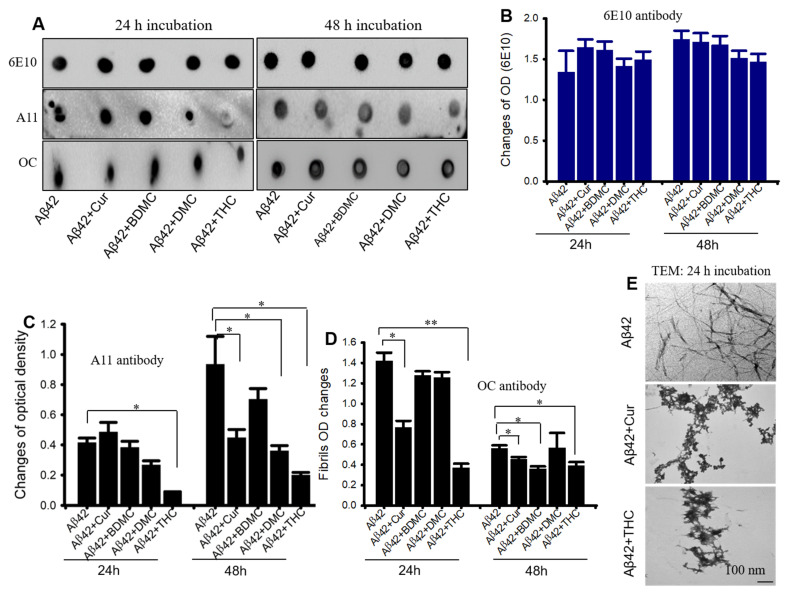
THC showed greater Aβ42 aggregation inhibition than other Cur derivatives. Aβ42 peptide was disaggregated with HFIP and dissolved in 60 mM NaOH and diluted with PBS (pH 7.4) and incubated for 24–48 h in presence or absence of different Cur-derivatives. About 10 µL of peptide was spotted on nitrocellulose membrane and probed with Aβ-specific antibodies (6E10, A11 and OC). The blots were developed with chemiluminescent reagent and the optical density of each band was measured with ImageJ software. (**A**): Representative dot blot images after treatment with all the Cur-derivatives (1 µM) for 24–48 h. (**B**): No significant differences were observed with 6E10 antibodies. (**C**): Oligomer formation (A11 antibodies) was significantly reduced by Cur, THC and DMC but not by BDMC at 48 h only. (**D**): Cur and THC inhibited Aβ42 fibril formation (OC antibodies). Images are representative of two independent experiments. (**E**): Representative TEM images showed that both Cur and THC inhibited Aβ42 fibril formation. Scale bar 100 nm and is applicable to all images. * *p* < 0.05 and ** *p* < 0.01 compared to Aβ42.

**Figure 6 antioxidants-10-01592-f006:**
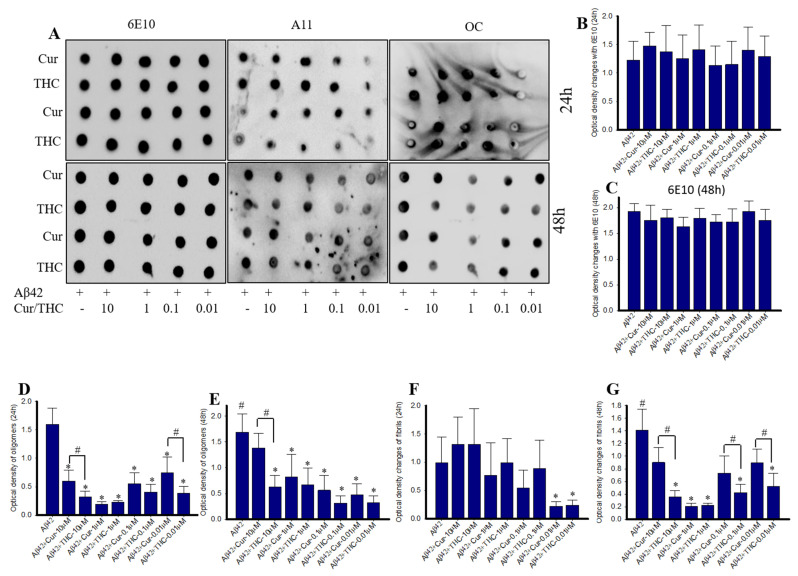
Lower concentrations of THC and Cur inhibited Aβ42 oligomers and fibrils. Aβ42 peptide (40 µM) was disaggregated with HFIP and dissolved in 60 mM NaOH and diluted with PBS (pH 7.4) and incubated for 24–48 h in presence or absence of different Cur-derivatives (1 µM). In this case, 10 microliters of peptide from each group were spotted on nitrocellulose membranes and probed with 6E10, A11 and OC antibodies. (**A**): Representative dot blot images image after 24- and 48-h of Aβ42 treated with different concentrations of Cur and THC (in µM: 10, 1, 0.1 and 0.001). (**B**,**C**): Optical density of blots probed with 6E10. No significant differences were observed among the groups. (**D**,**E**): Aβ42 oligomer formation (A11 antibody) was inhibited by Cur and THC treatment at all the concentrations. THC showed greater inhibition than Cur. (**F**,**G**): Aβ42 fibril formation was significantly inhibited by both Cur and THC and lower concentrations of THC showed greater inhibition of fibrils. * *p* < 0.05 in comparison untreated group; # *p* < 0.05 in comparison to Cur-treated groups.

**Figure 7 antioxidants-10-01592-f007:**
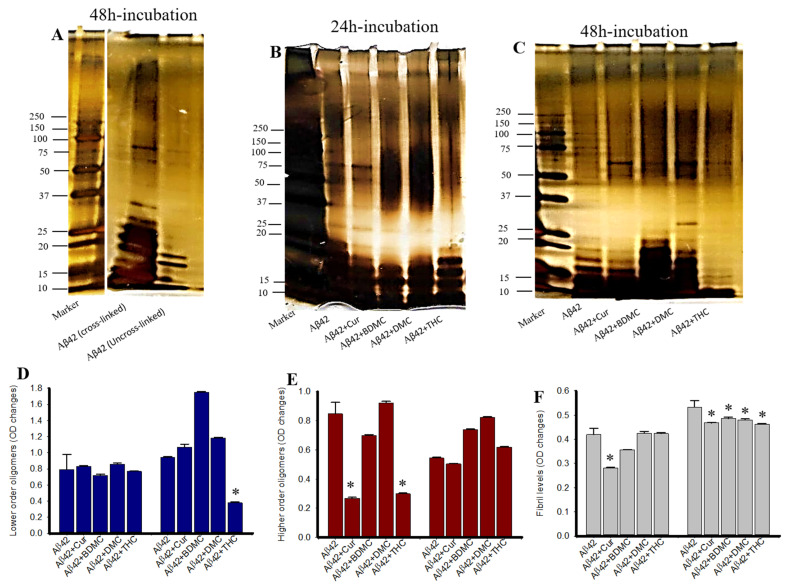
Photo-induced cross-linking of unmodified protein after treatment of different concentrations of Cur and THC. Synthesized Aβ42 was treated with HFIP and then diluted with 60 mM NaOH, 40% deionized water and 50% 20 mM sodium phosphate buffer and incubated with or without different Cur/turmeric derivatives (1 µM) for 24–48 h. The PICUP was then performed and the peptide was run in SDS-PAGE, followed by silver staining. (**A**): Silver-stained Aβ42 gel image with and without cross-linking. Note that without cross-linking, there is no oligomer or fibril formation observed. (**B**,**C**): Representative silver-stained images of Aβ42 with PICUP after treatment with different Cur/turmeric derivatives. (**D**–**F**): Oligomer formation was significantly decreased by Cur and THC. * *p* < 0.05 compared to Aβ42 untreated group.

**Figure 8 antioxidants-10-01592-f008:**
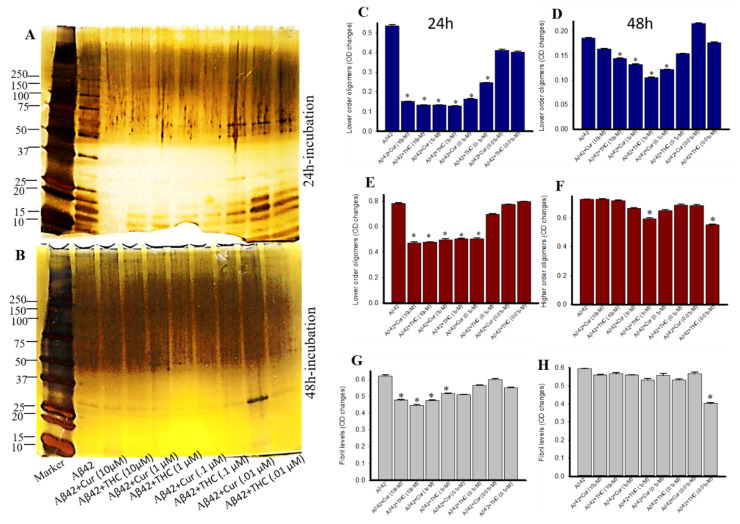
Lower concentration of Cur and THC inhibited Aβ42 aggregation. Synthesized Aβ42 was treated with HFIP and then diluted with 60 mM NaOH, 40% deionized water and 50% 20 mM sodium phosphate buffer and incubated with or without different concentrations of Cur and THC (in µM: 10, 1, 0.1 and 0.001) for 24–48 h. (**A**,**B**): Silver-stained Aβ42 peptide after treatment with different concentrations of Cur and THC, followed by PICUP reaction. (**C**–**H**): Lower order, higher order oligomers and fibrils were significantly inhibited by both Cur and THC. * *p* < 0.05 compared to the Aβ42 untreated group.

**Figure 9 antioxidants-10-01592-f009:**
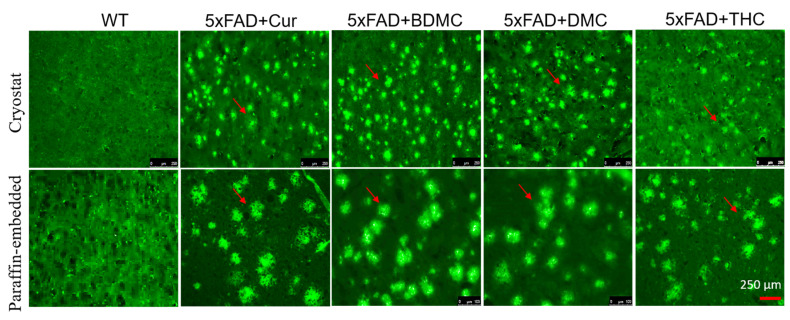
Curcumin/turmeric derivatives labeled Aβ plaques similar to Cur in vitro. 5× FAD brain tissue was sectioned by cryostat (40 µm) and paraffin embedded sections (5 µm) were stained with Cur, BDMC, DMC, THC and the images were taken under fluorescent microscope using appropriate excitation and emission filters. Note that all Cur derivatives labeled Aβ plaques in the cortical tissue of 5× FAD mice. Arrows indicate Aβ-plaques. Scale bar indicates 250 µm and is applicable to all images.

**Figure 10 antioxidants-10-01592-f010:**
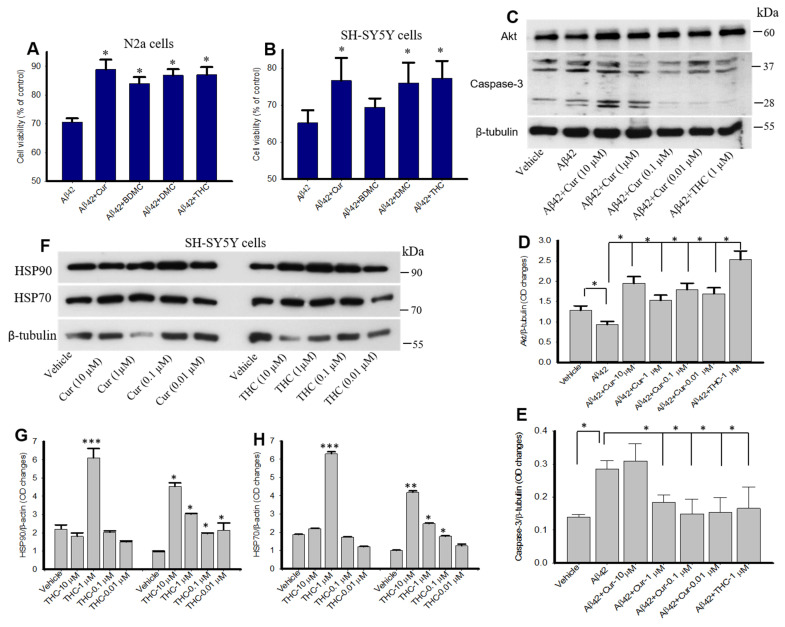
Neuroprotective effects of Cur and Cur/turmeric derivatives in vitro. (**A**,**B**): N2a (A) and SH-SY5Y (B) cells were treated with Aβ42 (10 µM) in presence and absence of Cur-derivatives (1 µM) for 24 h and cell viability was performed by MTT assay. All Cur/turmeric derivatives that were tested protected against Aβ42-induced neurotoxicity in both the cell lines. (**C**–**E**): Western blots of Akt and caspase-3 levels in SH-SY5Y cells after treatment with Aβ42 (10 µM) for 24 h in presence and absence of different concentrations of Cur and THC. The caspase-3 were significantly reduced by different concentrations of Cur and THC (1 µM). (**F**–**H**): SH-SY5Y cells were treated with different concentrations of Cur and THC. Treatment of THC induces HSP90 and HSP70 in SH-SY5Y cells similar to Cur-treated cells. * *p* < 0.05, ** *p* <0.01 and *** *p* < 0.001compared with Aβ42.

**Table 1 antioxidants-10-01592-t001:** Binding free energy of Aβ40 and Aβ42 with KCur and THC. The binding affinity were calculated using MMPBSA method. All energy values are in kcal/mol. The values are mean ± standard deviation. Here E_VDW_ = van der Waals energy, E_EEL_ = electrostatic energy, and E_GB_ = polar and E_SURF_ = non-polar contribution to the solvation energy, H_Tot_ = the enthalpy of binding.

Components Error	Aβ40 + KCur	Aβ40 + THC	Aβ42 + KCur	Aβ42 + THC
ΔE_VDW_	−24.8503 ± 1.6431	−31.9872 ± 1.9511	−34.9977 ± 1.9897	−31.5872 ± 1.8513
ΔE_EEL_	−5.2240 ± 2.7017	−8.7495 ± 3.5774	2.5387 ± 0.9428	−10.7495 ± 3.9772
ΔE_GB_	13.4585 ± 2.6444	18.9348 ± 3.4770	11.8059± 1.2426	19.3348 ± 3.5790
ΔE_SURF_	−3.8720 ± 0.2205	−4.6870 ± 0.3075	−4.7245 ± 0.2368	−4.3870 ± 0.3305
ΔH_Tot_	−20.4879 ± 1.4694	−26.4890 ± 1.9629	−30.4550 ± 1.6558	−27.3890 ± 1.7620

## Data Availability

All the data analyzed for this manuscript are included. The analyzed raw data are available upon reasonable request to the corresponding author. The data are not publicly available due to involvement of multiinstitutional researchers.

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
