# Peer review of "Tetrahydrocurcumin Has Similar Anti-Amyloid Properties as Curcumin:  In Vitro Comparative Structure-Activity Studies"

_antioxidants, 2021, doi:10.3390/antiox10101592_

Round 1

Reviewer 1 Report

The manuscript by Panchanan Maiti et al describes  anti-amyloid properties of few  curcumin derivatives, namely  bisdemethoxycurcumin (BDMC), demethoxycurcumin  (DMC) and Tetrahydrocurcumin (THC) with the aim to identify the most promising inhibitor of amyloid beta protein (Aβ) aggregation. The study was carried out using molecular docking and molecular dynamics (MD) simulations in combination with in vitro and in vivo assays. The topic is interesting however computational methodology is not applied in a way that corresponds to state-of-the-art. They have tried to analyze protein-ligand interactions but some basic experiments are needed before considering the manuscript for publication.

  1. Molecular docking protocol has to be validated
  2. Docking and scoring can be used to predict binding modes and discriminate between binders and non-binders, but are not particularly accurate in determining ligand-binding affinities. I suggest to use MM/PBSA and MM/GBSA methods to estimate the free energy of binding.
  • MD simulations are rarely performed singly in recent years. I suggest to run simulations in triplicate.

Minor points:

  • Figure 1 and Figure 2 are not clear. Molecular structures in Figure 1B should be properly shown.
  • Line 410: “integrated”, please correct
  • Table 2 appears as the output of docking calculations
  • Line 448: “covalent” please explain
  • On the basis of Tables 2 and 4 please explain the rational in selecting KCur and THC for MD simulations.

Author Response

Reviewer-1

The manuscript by Panchanan Maiti et al describes  anti-amyloid properties of few  curcumin derivatives, namely  bisdemethoxycurcumin (BDMC), demethoxycurcumin  (DMC) and Tetrahydrocurcumin (THC) with the aim to identify the most promising inhibitor of amyloid beta protein (Aβ) aggregation. The study was carried out using molecular docking and molecular dynamics (MD) simulations in combination with in vitro and in vivo assays. The topic is interesting however computational methodology is not applied in a way that corresponds to state-of-the-art. They have tried to analyze protein-ligand interactions, but some basic experiments are needed before considering the manuscript for publication.

  1. Molecular docking protocol has to be validated

Response: We appreciate the reviewer’s suggestion. We have validated our molecular docking of curcumin derivatives against the Aβ42 and Aβ40 by comparing our docking results with that of the Kumar Garav and co-workers (Kumar, G.; et al. World Journal of Pharmaceutical Research, 2014, 3 (2), 2987-2999) ,Jakubowski, J. M. and co-workers (Jakubowski, J. M.; et al. Journal of Chemical Information and Modeling 2020, 60, 289-305), and Rizwan Hasan Khan and co-workers (Khan, R. H.; et al. International Journal of Biological Molecules 2019, 127, 250-270). Our results are in concert with those reports as curcumin derivatives, in our experiment, were observed to bind to the first helical region (8-25) of Aβ40 as well as Aβ42 with His13 and His14 forming hydrogen bonding with most of the curcumin derivatives. Additionally, Val 12, Lys 16, Leu 17, and Phe 20 were either involved in hydrogen bonding interaction or located in close vicinity.

Docking and scoring can be used to predict binding modes and discriminate between binders and non-binders but are not particularly accurate in determining ligand-binding affinities. I suggest using MM/PBSA and MM/GBSA methods to estimate the free energy of binding.

Response: We are thankful to the reviewer for the valuable suggestions. We have performed the MMPBSA calculations and the results has been updated as Table 4 in the revised manuscript (Please see page 11).

  1. MD simulations are rarely performed singly in recent years. I suggest running simulations in triplicate.

Response: We agree with reviewer point. To comply reviewer’s suggestions, we performed two more replicas for each complex. Thereafter we seamed all trajectory to a single MD trajectory and calculated the most populated trajectory using clustering methods. The results are shown in the revised manuscript are for the most populated trajectory which is statistically more accurate way to present the MD results. This information has been added to the method section. Please see page 3 and page 11 (Revised Fig 4) in the revised manuscript.

Minor points:

  • Figure 1 and Figure 2 are not clear. Molecular structures in Figure 1B should be properly shown.

Response: In Fig 1 and 2 where are trying to show the amino acids of Aβ40/42 involved in interaction with curcumin derivatives. We have updated the Fig 1B the revised manuscript. Please see page 8 in the revised manuscript.

  • Line 410: “integrated”, please correct

Response: It is corrected. Please see page no: 9 in the revised manuscript

  • Table 2 appears as the output of docking calculations

Response: We have deleted the Table 2 from the revised manuscript. Please see revised Tables in the revised manuscript.

  • Line 448: “covalent” please explain

Response: We changed the word “covalent” to “hydrophobic”, van der Waals, electrostatic interactions. Please see page 11 in the revised manuscript.

  • On the basis of Tables 2 and 4 please explain the rational in selecting KCur and THC for MD simulations.

Response: Previous studies reported that keto form of curcumin (Kcur) is strongly binds with Aβ, especially Aβ oligomers (Please see: Biomaterials, 2021 Mar;270:120686). We know that the keto form of curcumin has less stability in body fluids with limited bioavailable. Whereas, tetrahydrocurcumin (THC) is apparently more stable Cur-derivatives in most of the body fluids. Therefore, we were interested whether THC have similar interaction capability like keto curcumin with Aβ, we compared THC with KCur.

Reviewer 2 Report

The manuscript by Panchanan Maiti et al shows anti-amyloid and neuroprotective properties of some curcumin derivatives, by performing molecular docking/dynamics studies, PICUP assay, TEM analysis and by using cell and animal model for AD. In the second part the Authors perform in vitro and in vivo experiments and here I would like to make some considerations. Figure 5. the DOT analysis shown in Figure 5 is not very convincing as well as the densitometric analysis. The authors say that they did it twice, perhaps it would be appropriate to repeat it in order to obtain better and more representative data. Moreover, after 24H there are already many fibrils and the A11 antibodies should recognize oligomeric structures, perhaps an analysis with a different technique, for example ThT assay, would be useful to confirm and validate the data.Of course, all checks must be made to ensure that there is no interference between substances and probes or antibodies. Figure 6. Even in these case the dot blots are of poor quality Figure7- 8. The silver stained gels are of poor quality. I understand that the PICUP assay is rather difficult to perform but these images are not presentable. I don't understand the meaning of Fig.9 Figure 10. the housekeeping (beta-actin) should be the same in all lines

Author Response

Reviewer-2

The manuscript by Panchanan Maiti et al shows anti-amyloid and neuroprotective properties of some curcumin derivatives, by performing molecular docking/dynamics studies, PICUP assay, TEM analysis and by using cell and animal model for AD. In the second part the Authors perform in vitro and in vivo experiments and here I would like to make some considerations.

Figure 5. the DOT analysis shown in Figure 5 is not very convincing as well as the densitometric analysis. The authors say that they did it twice, perhaps it would be appropriate to repeat it in order to obtain better and more representative data.

Response: We agree with reviewer the data presented in Fig 5, by itself, would not be great, but it does provide some support for the other measures. In fact, we have replaced the image in the Fig 5, with a new one, which also supports our findings (please see page 12).

Moreover, after 24H there are already many fibrils and the A11 antibodies should recognize oligomeric structures, perhaps an analysis with a different technique, for example ThT assay, would be useful to confirm and validate the data. Of course, all checks must be made to ensure that there is no interference between substances and probes or antibodies.

Response: We agree with reviewer’s point about the A11 signal. We replaced A11 signals in the Fig 5 of the revised manuscript (please see page 12). We also agree with reviewer that the ThT assay will be better to monitor the Aβ aggregation kinetics. However, as curcumin and its derivatives also have similar excitation/emission wavelengths (420-450 nM), it would be too difficult to monitor the Aβ aggregation with the ThT assay after treatment with curcumin-derivatives.

Figure 6. Even in these cases the dot blots are of poor quality Figure7- 8. The silver stained gels are of poor quality. I understand that the PICUP assay is rather difficult to perform but these images are not presentable.

Response: We agree with reviewer the Fig 7-8 might not be the best quality of image. We performed PICUP experiment few times and the best images were presented in the manuscript.

I don't understand the meaning of Fig.9 Figure 10.

Response: Fig 9 is showing that all the curcumin derivatives bind with Aβ plaques in the brain tissue of AD mouse model. We did not find any differences in Aβ labelling capabilities of different Cur-derivatives. Fig 10 showed that tetrahydrocurcumin was also capable to inhibit Aβ-induced neurotoxicity in vitro, similar to curcumin and other curcumin-derivatives.  

The housekeeping (beta-actin) should be the same in all lines.

Response: It was our typo error. It should be β-tubulin. We have corrected this in the revised manuscript. Please see the revised the Fig 10.

Reviewer 3 Report

The article „Tetrahydrocurcumin has similar anti-amyloid properties as curcumin: an in vitro comparative structure-activity studies “(Jang et al.) tested several derivatives of curcumin on their potency against b-amyloid-induced toxicity which might be an option for the treatment of AD. Due the low bioavailability of curcumin, they focused on the neuroprotective properties of the more stable tetrahydrocurcumin. By applying biophysical methods, electron microscopy and N2a, CHO and SH-SY5Y cells using Aβ42 (10 μM) as a toxin and tissue derived from the 5xFAD mouse model of AD, they found that all derivatives tested showed a similar degree of neuroprotection in vitro and ex vivo, whereby tetrahydrocurcumin showed preferable anti-amyloid properties than other derivatives.

The pursuit in searching bioavailable small molecules interfering with Ab aggregation to toxic oligomers is unfortunately still an unmet medical need in the developing of treatment options against AD. As such, the characterization of molecular and biophysical as well as neuroprotective properties of curcumin and derivatives is reasonable. The MS is well written and the experiments well performed. However, some comments from my side:

Major

  1. The authors were using the standard b-amyloid isoforms Ab1-40 and Ab1-42. However, also other isoforms are gaining more and more attention, such as pyroglutamate-modified Ab (AbpE3) and nitrated Ab (3NTyr10-Ab). Since the authors did not conduct experiments with these isoforms, irrespective of the abundance in AD, they should at least discuss those isoforms in their MS, e.g, AbpE3 serves as a seed for Ab1-42 aggregation and might change the binding properties of curcumin and hence oligomerisation.
  2. The constraints of the biophysical methods used and TEM do not allow the application of physiological Ab concentrations in the nM range. However, this does not apply for those experiments performed with cell systems. Here it is obviously feasible using lower concentrations as 10µM. I would suggest repeating the experiments in N2a, CHO and SH-SY5Y cell with Ab1-42 (100nM) and the respective stochiometric ratio with the lowest, effective concentration of curcumin and derivatives.

Minor

  1. Please check all acronyms and its first appearances.
  2. Delete “Materials and Methods” at the end of your introduction
  3. In some figures, the font is distorted
  4. Check units: µM was used instead of µm

Author Response

Reviewer-3

The article “Tetrahydrocurcumin has similar anti-amyloid properties as curcumin: an in vitro comparative structure-activity studies “(Jang et al.) tested several derivatives of curcumin on their potency against b-amyloid-induced toxicity which might be an option for the treatment of AD. Due to the low bioavailability of curcumin, they focused on the neuroprotective properties of the more stable tetrahydrocurcumin. By applying biophysical methods, electron microscopy and N2a, CHO and SH-SY5Y cells using Aβ42 (10 μM) as a toxin and tissue derived from the 5xFAD mouse model of AD, they found that all derivatives tested showed a similar degree of neuroprotection in vitro and ex vivo, whereby tetrahydrocurcumin showed preferable anti-amyloid properties than other derivatives. The pursuit in searching bioavailable small molecules interfering with Ab aggregation to toxic oligomers is unfortunately still an unmet medical need in the development of treatment options against AD. As such, the characterization of molecular and biophysical as well as neuroprotective properties of curcumin and derivatives is reasonable. The MS is well written, and the experiments well performed. However, some comments from my side:

Major

  1. The authors were using the standard b-amyloid isoforms Ab1-40 and Ab1-42. However, also other isoforms are gaining more and more attention, such as pyroglutamate-modified Ab (AbpE3) and nitrated Ab (3NTyr10-Ab). Since the authors did not conduct experiments with these isoforms, irrespective of the abundance in AD, they should at least discuss those isoforms in their MS, e.g, AbpE3 serves as a seed for Ab1-42 aggregation and might change the binding properties of curcumin and hence oligomerization.

Response: Thanks for reviewer’s valuable suggestion. We have added some sentences about AbpE3 serves as a seed for Ab1-42 aggregation in the revised manuscript. Please page no 19, para 3.

  1. The constraints of the biophysical methods used and TEM do not allow the application of physiological Ab concentrations in the nM range. However, this does not apply for those experiments performed with cell systems. Here it is obviously feasible using lower concentrations as 10µM. I would suggest repeating the experiments in N2a, CHO and SH-SY5Y cells with Ab1-42 (100 nM) and the respective stoichiometric ratio with the lowest, effective concentration of curcumin and derivatives.

Response: We agree with reviewer’s point about performing a set of experiments with 100 nM concentration of Ab1-42. However, in our previous experiment, with low concentration of Ab1-42, like 100 nM or 1 µM, we did find 5-10% cell death. In order to show neuroprotective effect of Cur-derivative, we need to get an optimum cell death (may be 30-40%) and with several experiment we have shown that 10 uM of Ab1-42 showed 30-40% of the cells died, therefore, we performed all of our neurotoxicity assays using a 10-µM concentration of Ab1-42. These descriptions are added in the revised manuscript (please see page 5).

Minor

  1. Please check all acronyms and their first appearances.

Response: All the acronyms are checked in the revised manuscript

  1. Delete “Materials and Methods” at the end of your introduction

Response:  The words “Materials and Methods” is deleted from the revised manuscript. Please see page no 3 in the revised manuscript.

  1. In some figures, the font is distorted

Response: The “font” of all the figures are corrected in the revised manuscript

  1. Check units: µM was used instead of µm

Response: It is corrected in the revised manuscript. Please see page 16.

Round 2

Reviewer 1 Report

The authors properly answered my questions in their response letter. There is an aspect however which has to be clarified.  They write: “We performed two more replicas for each complex. Thereafter we seamed all trajectory to a single MD trajectory and calculated the most populated trajectory using clustering methods. The results are shown in the revised manuscript are for the most populated trajectory which is statistically more accurate way to present the MD results.” It is not clear which trajectory Fig.3 refers to, what’s the difference with Fig 3 of the previous version of the manuscript?

Author Response

Reviewer 1

The authors properly answered my questions in their response letter. There is an aspect however which has to be clarified.  They write: “We performed two more replicas for each complex. Thereafter we seamed all trajectory to a single MD trajectory and calculated the most populated trajectory using clustering methods. The results shown in the revised manuscript are for the most populated trajectory which is a statistically more accurate way to present the MD results.” It is not clear which trajectory Fig.3 refers to, what’s the difference with Fig 3 of the previous version of the manuscript?

Response: We are thankful to the reviewer for acknowledging our revision in the revised manuscript. We apologize for the confusion in the method section. We have revised the statement as “The binding energy by MMPBSA and the final snapshots shown in Figure 4 are calculated for the most populated trajectory which is a statistically more accurate way to present the MD results.” Please see page 4 in the revised manuscript

Furthermore, we have corrected the caption in Figure 3 as “ RMSD and RMSF are shown for the first 50 ns of the simulations.” Please see page 10 in the revised manuscript.

Reviewer 2 Report

The authors suitably replied my questions. 

Author Response

Thank you very much for your critical review which helps a lot to improve our manuscript.

Reviewer 3 Report

I’m fine with the author’s responses

Author Response

Thank you very much for your valuable suggestions to improve our manuscript.